# Divergent Specialization of Simple Venom Gene Profiles among Rear-Fanged Snake Genera (*Helicops* and *Leptodeira*, Dipsadinae, Colubridae)

**DOI:** 10.3390/toxins14070489

**Published:** 2022-07-15

**Authors:** Peter A. Cerda, Jenna M. Crowe-Riddell, Deise J. P. Gonçalves, Drew A. Larson, Thomas F. Duda, Alison R. Davis Rabosky

**Affiliations:** 1Ecology and Evolutionary Biology, University of Michigan, Ann Arbor, MI 48109, USA; j.crowe-riddell@latrobe.edu.au (J.M.C.-R.); deisejpg@umich.edu (D.J.P.G.); larsonda@umich.edu (D.A.L.); tfduda@umich.edu (T.F.D.J.); ardr@umich.edu (A.R.D.R.); 2Museum of Zoology, University of Michigan, Ann Arbor, MI 48108, USA; 3School of Agriculture, Biomedicine and Environment, La Trobe University, Melbourne, VIC 3086, Australia; 4Department of Biology, Indiana University, Bloomington, IN 47405, USA

**Keywords:** transcriptomics, gene expression, snake venom, opisthoglyphous, C-type lectin, metalloproteinase, toxin variation, neotropics, Peruvian Amazon, Central America

## Abstract

Many venomous animals express toxins that show extraordinary levels of variation both within and among species. In snakes, most studies of venom variation focus on front-fanged species in the families Viperidae and Elapidae, even though rear-fanged snakes in other families vary along the same ecological axes important to venom evolution. Here we characterized venom gland transcriptomes from 19 snakes across two dipsadine rear-fanged genera (*Leptodeira* and *Helicops*, Colubridae) and two front-fanged genera (*Bothrops*, Viperidae; *Micrurus*, Elapidae). We compared patterns of composition, variation, and diversity in venom transcripts within and among all four genera. Venom gland transcriptomes of rear-fanged *Helicops* and *Leptodeira* and front-fanged *Micrurus* are each dominated by expression of single toxin families (C-type lectins, snake venom metalloproteinase, and phospholipase A2, respectively), unlike highly diverse front-fanged *Bothrops* venoms. In addition, expression patterns of congeners are much more similar to each other than they are to species from other genera. These results illustrate the repeatability of simple venom profiles in rear-fanged snakes and the potential for relatively constrained venom composition within genera.

## 1. Introduction

Many animal groups use venoms that are comprised of toxic compounds to subdue prey and defend against predators [1]. Venom composition tends to differ considerably among closely related taxa within these groups, possibly due to differences in the type and diversity of prey species [2,3,4,5,6,7,8]. While venoms of a variety of taxa have been characterized [1,4,9,10], the venom composition of members of related but understudied groups is important for reconstructing the evolution of these complex phenotypes across taxa.

Much of our knowledge of snake venoms is based on information from front-fanged species of the families Viperidae (e.g., rattlesnakes) and Elapidae (e.g., coral snakes) [6,11,12,13]. These snakes inject venom through hypodermic needle-like fangs that are either hinged (viperids) or fixed (elapids) at the front of the mouth (Figure 1B) and utilize a high-pressure venom delivery system (Figure 1A) [14]. While most venom components can be found throughout venomous snake species, the venom composition of these two families tends to be drastically different [14]. Viperid venoms are typically hemorrhagic or cytotoxic and largely contain snake venom metalloproteinases (SVMPs), snake venom serine proteases (SVSPs), and phospholipase A2s (PLA2s) [15]. On the other hand, venoms of elapids are usually neurotoxic and primarily contain either PLA2s, three-finger toxins (3FTxs), or a combination of these two types [15]. While these toxin families dominate the venom profiles of front-fanged snakes, other toxins, such as C-type lectins (CTLs), are also found at varying levels [15]. Front-fanged species can differ in venom complexity [16], ranging from “simple” venoms comprised mostly of one toxin family to “complex” venoms comprised of many toxin families [6] that may be associated with inter- and intraspecific differences in predator–prey interactions [2,5,6,17].

Less of our knowledge on snake venoms comes from rear-fanged snakes of the family Colubridae (*sensu* Pyron et al. [18]), which makes up approximately half of all snake species [19]. Venomous colubrids produce toxins in the oral Duvernoy’s gland (hereafter referred to as the “venom gland”; Figure 1A), which are delivered through grooved or ungrooved fangs located at the back of the mouth (Figure 1B) via a low-pressure venom delivery system [19,20]. Although the family contains approximately 700 venom-producing paraphyletic rear-fanged species [14], venoms have only been investigated in a few of them (Table 1). Emerging trends seem to suggest that Colubrid venoms tend to have relatively simple compositions and that the subfamilies appear to follow a compositional dichotomy [21]. Venoms of the dipsadine subfamily tend to be dominated by SVMPs, such as some viperids, while venoms of the colubrine subfamily largely contain 3FTxs, such as some elapids (Table 1 and references within). Variation in the venoms of colubrid snakes has not been well studied, but there is evidence of ontogenetic shifts in venom composition of *Boigia irregularis* [22], as well as some geographic variation in the venom composition of *Tantilla nigriceps* [23]. Nonetheless, the venom compositions of rear-fanged snakes are still largely unknown, limiting our ability to both describe general patterns in the diversity of colubrid venoms as well as accurately model the evolutionary dynamics of snake venoms more broadly.

To increase our understanding of the venom composition of rear-fanged snakes, we characterized venom gland transcriptomes of members of the dipsadine genera *Helicops* and *Leptodeira*. While venoms of members of these genera have not previously been examined through RNA sequencing approaches, they have been subject to functional studies, and so some inferences about their composition are available [32,33,34,35,36,37]. For example, *Leptodeira annulata* and its subspecies *Leptodeira annulata pulchriceps* have venoms that differ in terms of serine protease activity, but both appear to be rich in SVMPs and PLA2s as they exhibit proteolytic, hemorrhagic, and neurotoxic activities [33,34,35]. The venom of *Helicops angulatus* exhibits neurotoxic but not hemorrhagic activity and contains a cysteine-rich secretory protein, termed helicopsin, that causes respiratory paralysis in mice [36,37]. The absence of hemorrhagic activity implies that venoms of *H. angulatus* do not contain SVMPs, as these metalloproteases tend to induce hemorrhaging [38]. Hence, not all dipsadine snakes apparently possess a viperid-like venom, and some members of this subfamily may instead contain members of other toxin types.

We provide a first characterization of the venom gland transcriptomes of members of the dipsadine genera, *Helicops* and *Leptodeira*, and compare them with venom gland transcriptomes of front-fanged snakes of the genera *Bothrops* (Viperidae) and *Micrurus* (Elapidae). To do this, we sequenced venom gland transcriptomes from 14 species of these four genera, including multiple individuals of four species. We determined (i) which major toxin families are expressed, (ii) whether these venoms are simple or complex, and (iii) inter- and intraspecific levels of variation in venom composition among members of these genera. Although the venom profiles of *Bothrops* and *Micrurus* species have been characterized previously, we generated new sequence data for these genera to enable effective comparisons of transcriptomes generated via the same library preparation methods, sequencing approaches, and bioinformatic procedures.

## 2. Results

### 2.1. Venom Gland Gene Family Recovery

We produced and analyzed the venom gland transcriptomes of 19 individuals of six rear-fanged snake species (*Helicops n* = 5 and *Leptodeira n* = 4; Table 2) and eight front-fanged snake species (*Bothrops n* = 3 and *Micrurus n* = 8; Table 2). The number of paired reads recovered per sample ranged from 12,936,858 to 24,938,732 (Table 3). We found that the total toxin sequence count was associated with sequencing platforms used: transcriptomes that were sequenced on a HiSeq 4000 produced a higher toxin transcript count than those on a NovaSeq 6000, including transcriptomes of conspecifics or congeners. The number of unique toxin transcripts recovered from the venom gland of *Helicops* species ranged from 47 to 91 (Figure 1C). The most frequently identified toxin families in *Helicops* were snake venom metalloproteinase (SVMPs) and C-type lectins (CTLs). Several copies of cysteine-rich secretory proteins (CRiSPs) were recovered from two individuals of *H. angulatus* and one individual of *H. leopardinus* (Figure 1C). The number of unique toxins recovered from *Leptodeira* venom gland transcriptomes ranged from 29 to 99 (Figure 1D). Similar to the rear-fanged *Helicops*, the most common toxin families in *Leptodeira* were SVMPs, CTLs, and PLA2s (Figure 1D). The number of unique toxins recovered from *Micrurus* species ranged from 37 to 141. These largely included SVMPs, PLA2s, and CTLs transcripts (Figure 1E). The number of unique toxin transcripts recovered from *Bothrops* species ranged from 89 to 131, and the most common toxin families recovered were SVMPs, snake venom serine proteinases (SVSPs), CTLs, and PLA2s (Figure 1F).

### 2.2. Venom Gland Transcriptome Expression

In total, we found that total toxin expression encompassed between 17% and 91% of the total expression in the venom gland transcriptomes studied (Table 3). We found that CTLs were the most highly expressed toxin family of *Helicops* and comprised between 63 and 99% of all toxins expressed (Figure 2). SVMPs were the second most abundant toxin family expressed (0.6–36%) in all but one individual of *H. angulatus*, for which CRiSPs toxins made up 24% of the venom expression profile (Figure 2). We found that SVMPs also comprised a considerable portion of the venom gland expression profile of rear-fanged *Leptodeira* species and encompassed 83–98% of the expressed toxins (Figure 2). The second most highly expressed toxins in *Leptodeira* were CTLs which ranged from 0.7 to 7% of expressed toxins (Figure 2). In front-fanged *Micrurus* species, we found that venom gland expression was dominated by one or two toxin families. All *Micrurus* species expressed PLA2s at a high level (49–99%; Figure 2). Kunitz-type serine protease inhibitors were the second most highly expressed toxin in both individuals of *M. lemniscatus* that we examined (22–26%; Figure 2), whereas 3FTxs were the second most highly expressed toxins of all other *Micrurus* species (13–21%; Figure 2). In front-fanged *Bothrops atrox* and *Bothrops brazili*, SVMPs had the highest expression, comprising 48 and 50% of the venom profile, respectively. (Figure 2). Bradykinin-potentiating peptides were the second most abundant toxin family found in *B. atrox* (28%), while CTLs were the second most abundant toxin found in *B. brazili* (20%; Figure 2). SVMPs and CTLs were expressed at similar levels (~30%) in *Bothrops bilineatus* (Figure 2).

### 2.3. Complexity and Variation

We calculated Shannon diversity indices to compare levels of venom complexity across individuals. These values ranged from a low of 0.662 for *Helicops leopardinus* to a high of 3.251 for *Bothrops brazili* (Figure 2). Overall, the four genera ranked from lowest to highest levels of venom complexity as follows: *Helicops* (1.289), *Micrurus* (1.602), *Leptodeira* (1.954), and *Bothrops* (3.014). We further estimated beta diversity statistics to determine levels of intra- and interspecific variation in venom composition. For the four species for which multiple individuals were examined, *L. annulata* and *M. lemniscatus* exhibited the lowest values (0.097 in both cases), while *H. angulatus* (0.239) and *M. obscurus* (0.497) had higher values. Within-genera comparisons showed that *Leptodeira* (0.097) and *Bothrops* (0.355) exhibited the lowest and highest values for genera, respectively, whereas *Helicops* (0.190) and *Micrurus* (0.180) had intermediate values. While intraspecific variation was found among members of these genera, we saw a greater similarity in venom gland transcriptome composition among individuals within genera than among genera (Figure 2).

## 3. Discussion

We used a transcriptomic approach to characterize and compare patterns of toxin variation in the venom gland transcriptomes of members of two rear-fanged (*Helicops* and *Leptodeira*) and two front-fanged (*Bothrops* and *Micrurus*) snake genera. While snake venom metalloproteinases (SVMPs) transcripts dominate the venom profile of the *Leptodeira* species, C-type lectins (CTLs) are the most highly expressed toxin family of *Helicops* species (Figure 2). Venom profiles of the front-fanged species are similar to those reported from these taxa previously [39,40,41]. While we were able to recover toxin transcripts from numerous toxin families across all individuals, we found that toxin expression within genera is typically dominated by only a single toxin class: CTLs in *Helicops*; phospholipase A2s (PLA2s) in *Micrurus*; and SVMPs in both *Leptodeira* and *Bothrops*. Shannon diversity indices suggest that while complexity may vary among individuals, differences may be due to the relative contribution of each underlying transcript rather than differences in toxin family abundance. We also observed low levels of variation among individuals of the same genera but high levels of variation across genera. 

The dipsadine rear-fanged species that we examined mostly exhibited low levels of venom complexity, given that their venom gland expression profiles were largely composed of transcripts of only single toxin classes (Figure 2). In line with previous results, front-fanged snakes tended to exhibit higher levels of venom complexity. Increased venom complexity has been correlated with large dietary breadth in both venomous cone snails and North American pit vipers [4,6]. The venoms of several rear-fanged snakes are known to have a “simpler” venom than their front-fanged relatives [21,26,27,42,43]. The lack of complexity observed for rear-fanged snakes may be due to the highly specialized diets that many colubrid snakes exhibit [14]. Broadly, *Leptodeira* appear to specialize in frogs while *Helicops* specialize in fish, though a formal analysis of dietary specialization in these two genera has not been performed [44,45]. However, the toxicological diversity of specific toxin families from the species described here has not been determined. The investigation of toxin function may reveal physiological targets and functions specific to prey types, which have been found in neurotoxins described from several rear-fanged snakes [22,26].

Several members of the colubrid subfamily Dipsadinae have been shown to use a hemorrhagic venom that is largely comprised of SVMPs [21,42]. The *Leptodeira* species we examined exhibited this type of venom profile. However, *Helicops* species had venom gland expression profiles that are dominated by transcripts of a single toxin family, CTLs. No previous studies have shown snakes that produce venom dominated by CTLs. While functional attributes of CTLs of rear-fanged snakes are not known, CTLs of front-fanged snakes are multifunctional heterodimers that affect hemostasis by acting as anticoagulants, which can cause hemorrhaging, or as procoagulants, which can cause blood clotting [46]. In addition, CTLs of front-fanged snakes have been shown to evolve rapidly [47]. Recently, Xie et al. [48] found extensive duplication of novel dimeric CTLs genes unique to *H. leopardinus*. The novel CTLs found in *H. leopardinus* were shown to be under positive selection, but the distribution of these CTLs across the genus *Helicops* is currently unknown [48]. A previous study of *H. angulatus* found its venom lacked hemorrhagic activity, which is typically associated with SVMPs and CTLs [36]. Instead, its venom was shown to exhibit neurotoxic activity, likely due to the presence of a previously uncharacterized cysteine-rich secretory protein (CRiSP) named helicopsin [36]. Of the *H. angulatus* individuals examined here, only two possess CRiSP transcripts, and only one expressed CRiSP at a considerable level (Figure 2). It is unknown if the CRiSP transcripts found in our Peruvian individuals are similar to helicopsin that was previously isolated from an individual of this species from Venezuela [36].

Our sampling included multiple individuals of some species and multiple species of four genera to evaluate patterns of intra- and interspecific variation of venom profiles as inferred from transcriptome data. The levels of variation within species differed considerably. Intraspecific variation of venom profiles of *L. annulata* and *M. lemniscatus* were low, while those of *H. angulatus* and *M. obscurus* were quite distinct. For example, individuals of *H. angulatus* differed in terms of the relative abundance of CRiSPs, SVMPs, and CTLs, despite being from the same locality. Further, the described absence of hemorrhagic activity of venoms of *H. angulatus* from Venezuela [37] suggests that this species exhibits geographic variation in venom. While venom gland profiles generated here generally match venom profiles that were described previously in other studies, geographic variation in venom composition may occur among some of the species surveyed here as well [12,40,41,49]. For example, while pooled venom of *B. brazili* from Pará, Brazil, contained 33% PLA2s, 27% SVMPs, and 14% SVSPs [49], SVMPs and CTLs were the predominant toxin components that are represented in the venom gland transcriptome of an individual from the Madre de Dios region of Peru. Geographic variation is rather common in predatory venomous species as populations presumably adapt to local prey assemblages [50,51]. However, caution should be taken when comparing venom gland transcriptomes and venom proteomes as there are cases in which there is a lack of correspondence between the abundance of expressed transcripts and translated proteins [40]. Further, the proportion of toxin expression in the whole transcriptome varied widely among the specimens examined (17.15–91.81%; Table 3). The variation in venom expression can possibly be attributed to differences in toxin expression over time [52]. For example, toxin expression is higher after feeding events to replenish the used venom proteins [52].

The two individuals of *M. obscurus* that we examined differed considerably in venom complexity, with the venom profile of one individual being nearly completely dominated by PLA2s, while that of the other individual was more complex and contained wapirins, 3FTxs, and Kunitz-type serine protease inhibitors (Figure 2; note that wapirins are included in the “Other” category in the figure). These two individuals differed in age, as one was a juvenile (*M. obscurus* 0665), while the other was an adult (*M. obscurus* 1054). Thus, the difference observed in *M. obscurus* may be due to an ontogenic shift from a more “complex” venom that is expressed by juveniles to a “simple” venom that is expressed by adults. Ontogenetic shifts in venom composition have been observed in many snake species, and this change is potentially due to differences in the diet as individuals age [17,22,53,54]. However, more intensive sampling is needed to determine if this is indeed the case in *M. obscurus*.

The species studied here displayed varying levels of interspecific variation in venom gland transcriptomes, which is a common pattern observed across venomous taxa [1,4,6,55]. Barua and Mikheyev [56] found that while many combinations of venom components were possible, different species tend to show similar venom profiles despite phylogenetic relatedness, albeit with different proportions of respective venom compositions. While interspecific variation was observed among species examined here, we do note that most of our taxa had venom gland profiles similar to those expected given their respective families [21,56], except for species of *Helicops*. It is not clear how or why the *Helicops* species examined here arrived at a potentially novel venom phenotype. The use of CTLs by *Helicops* species may be a more efficient strategy to capture their aquatic prey [44,45]. Proteomic and functional studies should be performed on *Helicops* venoms to determine the abundance of CTL proteins in these venoms and how they might be used in prey capture. Future exploration of colubrid venoms will help us further our understanding of how convergent and novel venom phenotypes have evolved across all venomous snakes.

## 4. Materials and Methods

### 4.1. Sampling

We collected individuals of *Bothrops*, *Helicops*, *Leptodeira*, and *Micrurus* species from multiple localities in Nicaragua and Peru between 2016 and 2018 (Table 2). Within ten minutes of euthanasia, we excised venom gland tissues and stored them in RNALater (Invitrogen, Carlsbad, CA, USA). While at remote locations, we stored samples at room temperature for up to three weeks during active collection expeditions. Then we stored the samples at −20 °C prior to export to the University of Michigan and subsequently at −80 °C until RNA extraction. We deposited whole voucher specimens in the Museo de Historia Natural, Universidad Nacional Mayor de San Marcos (MUSM) in Lima, Peru, and the University of Michigan Museum of Zoology (UMMZ; Table 2).

### 4.2. Extraction, Library Preparation, and Sequencing

We extracted the total RNA from venom glands using the PureLink RNA mini kit (Life Technologies, Carlsbad, CA, USA), following the recommended protocol for animal tissue. The total RNA was stored at −80 °C until submission to the University of Michigan’s Advanced Genomics Core, where the RNA was quantified with a Qubit fluorometer (Invitrogen, Carlsbad, CA, USA), and the size was visualized with an Aligent TapeStation (Santa Clara, CA, USA). Poly-A tail selected libraries were constructed with Illumina TruSeq RNASeq (San Diego, CA, USA) and NEBNext Ultra II RNA (Ipswich, MA, USA) library kits, and 150 bp paired-end sequencing was conducted on Illumina HiSeq4000 or Illumina NovaSeq6000 machines (Table 3).

### 4.3. Bioinformatics

We assessed the raw read quality for each individual using FastQC v0.11.6 [57] and used Trimmomatic v0.38 to trim adapters and remove low-quality reads [58] for downstream phylogenetic and transcriptomic analysis. We assembled the transcriptome of each individual using two methods, Trinity v2.6.6 [59] and Extender [60], a seed-and-extend assembler that has been shown to recover a high level of isoform diversity in snake venom families [61]. The extender assemblies were generated with combined trimmed forward and reverse paired reads that were merged with PEAR v0.9.6 [62].

We merged the assembled transcriptomes produced by Trinity and Extender to construct a single transcriptome for each individual. For each transcriptome, we followed a previously published protocol to filter out low-quality transcripts and chimeras [63,64] using transRate v1.0.3 [65] and a BLAST-based method [66], respectively. We used Corset v1.07 [67] to remove duplicate transcripts and select a single representative transcript (the longest) for each putative gene, using SALMON v0.11.2 [68] as our aligner option. To find open reading frames, we used transDecoder [69] and BLAST [70]; CD-HIT was used to reduce redundancy at 95% similarity [71]. We estimated RNA abundance (i.e., transcripts per million [TPM]) with the align_and_estimate_abundance.pl script in Trinity [59], which calls on RSEM v1.2.28 [72] and Bowtie2 v2.3.4.1 [73].

We created a custom database of non-toxin and toxin nucleotide sequences by downloading venom gland transcriptomes of *Crotalus adamanteus* [60], *Crotalus horridus* [74], *Micrurus fulvius* [75], *Boiga irregularis* [21], *Hypsiglena* sp. [21], and *Spilotes sulphureus* [26], the annotated genome of *Ophiophagus hannah* [76], and sequences of snake venom matrix metalloproteinase [77] from NCBI GenBank. A combination of these species has been used previously to identify putative rear-fanged snake venom components and represent a diversity of venomous snakes [26]. We used BLASTn [70] to identify known toxins in our transcriptomes using this custom database. We wrote a custom script in R [78] to annotate our nucleotide sequences, sort non-toxin and toxin transcripts by identifying annotations that matched known toxins from the custom database using the ‘grep’ function, and associate sequence identity with transcript abundance estimates. To determine the composition of toxin transcripts in the venom gland transcriptomes, we wrote a custom R script that counted the number of venom gene contigs and total TPM for each snake toxin gene family across individuals. Toxin sequences with a TPM of 0 were removed. We divided the total TPM of each toxin family over the total TPM of all toxin genes to give a proportion of each toxin family present in the venom gland transcriptome.

### 4.4. Assembly of Mitochondrial Sequences and Phylogenetic Tree Estimation

We assembled mitochondrial sequences for our samples using HybPiper v1.3.1 [79] and a target file consisting of sequences from 16 fully annotated, publicly available mitochondrial genomes downloaded from GenBank [80,81,82,83,84,85] (Appendix A). To prepare the target file, all sequences annotated as rRNA, tRNA, and protein-coding genes were extracted from the mitogenome using Geneious Prime 2020.2.3 (https://www.geneious.com (accessed on 5 January 2022)). We then manually curated sequence label formatting and combined all sequences into a single HybPiper target file. We processed the raw sequence reads for all of the transcrioptomes with trimmomatic [58] and used the options “ILLUMINACLIP:TruSeq3-PE-2.fa:2:30:10:2:TRUE SLIDINGWINDOW:5:20 LEADING:20 TRAILING:20 MINLEN:36” to trim adapters and reads with a PHRED score of less than 20. We spot-checked the trimmed reads with FASTQC [57]. We combined reads that became unpaired during trimming into a single file per sample and assembled target mitochondrial genes with HybPiper using forward, reverse, and unpaired reads and default settings. The HybPiper pipeline calls on Exonerate [86], BLAST+ [70], Biopython [87], BWA [88], SAMtools [89], GNU Parallel [90], and SPAdes [91].

The assemblies for protein coding, rRNA, and informative tRNA sequences were aligned with MAFFT v7.271 [92] and the options “–maxiterate 1000”, “–ep 0.123”, and “–genafpair”. Columns with >70% gaps were removed, and alignments were concatenated into a supermatrix with the pxclsq and pxcat commands in phyx, respectively [93]. We estimated a phylogenetic tree using IQ-TREE v 2.1.3 [94] and the options “–m TEST” and “–mset raxml” to determine the best-fitting model, which had 19 partitions, variously assigned GTR + F, GTR + F + I, GTR + F + G4, and GTR + F + I + G4 models. The maximum likelihood tree was visualized with Figtree v.1.4.4 (http://tree.bio.ed.ac.uk/software/figtree/ (accessed on 10 January 2022)).

### 4.5. Complexity and Variation

To estimate the levels of venom complexity, we calculated Shannon indices (*H′*) [95] based on each unique toxin and their respective TPM expressed in the transcriptomes of the individuals examined. For cases in which multiple individuals of a species were examined, we averaged the *H′* values from these individuals to estimate the venom complexity of the species. We also averaged the values across species to estimate the relative levels of venom complexity of the genera. We quantified the extent that samples differ in levels of intra- and interspecific variation of venom composition with calculations of pairwise proportional dissimilarity (PPD) values (i.e., Brays–Curtis distances, [96]) based on proportions of toxin families recovered. We used the PPD values to estimate and compare levels of intraspecific variation in venom composition for species from which multiple individuals were examined (i.e., *Helicops angulatus*, *Leptodeira annulata*, *Micrurus lemniscatus*, and *Micrurus obscurus*). We further averaged PPD values that were calculated among species of genera to evaluate levels of interspecific variation in venom composition.

### 4.6. diceCT and Segmentation

We used diffusible iodine contrast-enhanced computed tomography (diceCT) to scan a representative specimen from each genus using 1.25% Lugol’s iodine solution and a Nikon Metrology XTH 225ST microCTscanner (Xtect, Tring, UK) at the UMMZ, following protocols outlined in Callahan et al. [97]. We segmented the maxillary bone and venom gland using the ‘draw’ and ‘thresholding’ tools in Volume Graphics Studio Max version 3.2 (Volume Graphics, Heidelberg, Germany).

## Figures and Tables

**Figure 1 toxins-14-00489-f001:**
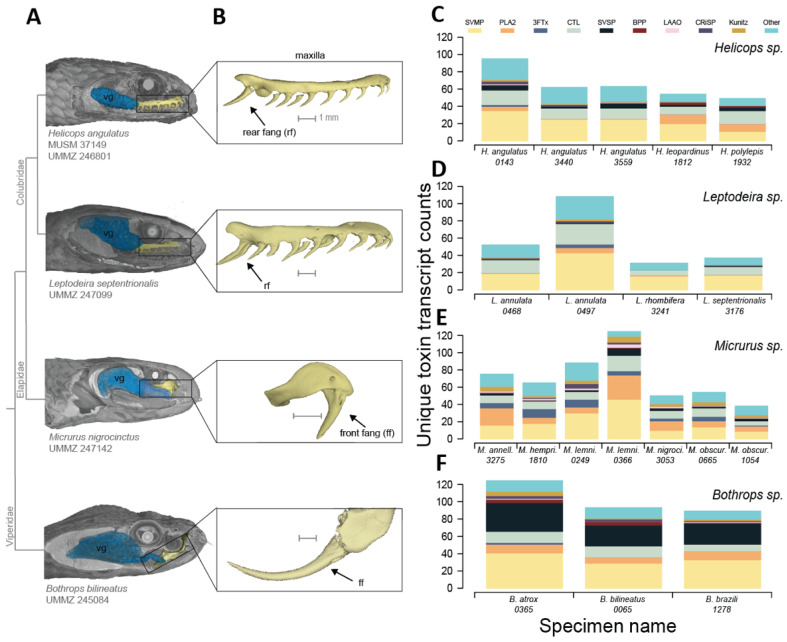
Representative venom delivery system for each genus and venom transcript counts of individuals. (**A**) Soft-tissue scan of the venom delivery system for each genus mapped onto a simplified phylogeny developed in this study. (**B**) Isolated maxilla bone from each representative genus showing position of fangs (arrow). (**C**) Unique toxin transcripts recovered from each *Helicops* individual. (**D**) Unique toxin transcripts recovered from each *Leptodeira* individual. (**E**) Unique toxin transcripts recovered from each *Micrurus* individual. (**F**) Unique toxin transcripts recovered from each *Bothrops* individual. MUSM = Museo de Historia Natural, Universidad Nacional Mayor de San Marco, UMMZ = University of Michigan Museum of Zoology, SVMP = snake venom metalloproteinase, PLA2 = phospholipase A2, 3FTx = three-finger toxin, CTL = C-type lectin, SVSP = snake venom serine proteinase, BPP = bradykinin-potentiating peptides, LAAO = L-amino acid oxidase, CRiSP = cystine rich secretory protein, Kunitz = Kunitz-type serine protease. Micro-CT scans of specimens vouchered in the University of Michigan Museum of Zoology (UMMZ) and Museo de Historia Natural de la Universidad Nacional Major de San Marcos (MUSM). We deposited these micro-CT scans used for venom and fang morphology for public access in the Morphosource ‘Scan All Snakes’ Project ID 00000C374 (https://www.morphosource.org/Detail/ProjectDetail/Show/project_id/374).

**Figure 2 toxins-14-00489-f002:**
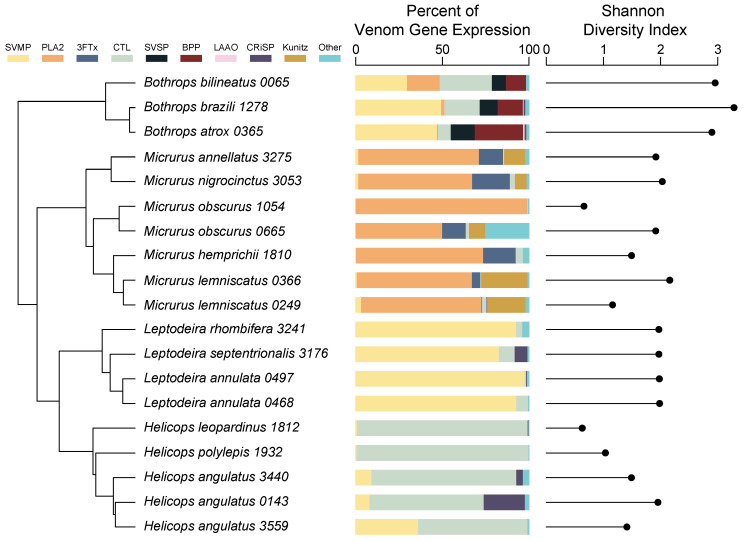
Expression of toxin gene families from venom gland transcriptomes (**center**) and overall venom transcriptome diversity (**right**) mapped to phylogeny inferred from mitochondrial gene sequences (**left**). Note that these data suggest there is generally higher similarity in venom profiles among individuals within genera than among genera for both metrics. *Bothrops* and *Micrurus* species are front-fanged, while *Leptodeira* and *Helicops* species are rear-fanged. SVMP = snake venom metalloproteinase, PLA2 = phospholipase A2, 3FTx = three-finger toxin, CTL = C-type lectin, SVSP = snake venom serine proteinase, BPP = bradykinin-potentiating peptides, LAAO = L-amino acid oxidase, CRiSP = cystine-rich secretory protein, Kunitz = Kunitz-type serine protease.

**Table 1 toxins-14-00489-t001:** Summary of venom gland transcriptome toxin profiles of several species of rear-fanged colubrid snakes. SVMP = snake venom metalloproteinase, 3FTx = three-finger toxin, CRiSP = cystine rich secretory protein, svMMP = snake venom matrix metalloproteinase, CTL = C-type lectins, Kunitz = Kunitz-type serine protease.

Subfamily	Species	Major Venom Component(s)	Reference
Colubrinae	*Ahaetulla prasina*	SVMPs, 3FTxs	[24]
	*Boiga irregularis*	3FTxs, SVMPs	[21]
	*Dispholidus typus*	SVMPs	[25]
	*Spilotes sulphureus*	3FTxs	[26]
	*Tantilla nigriceps*	3FTxs, CRiSPs, SVMPs	[23]
	*Trimorphodon quadruplex*	3FTxs, SVMPs	[27]
Dipsadinae	*Borikenophis portoricensis*	SVMPs	[24]
	*Conophis lineatus*	svMMPs	[28]
	*Hypsiglena* sp.	SVMPs, CRiSPs	[21]
	*Phalotris mertensi*	Kunitzs, SVMPs, CTLs	[29]
	*Philodryas olfersii*	SVMPs, CNPs	[30]
	*Thamnodynastes strigatus*	svMMPs	[31]

**Table 2 toxins-14-00489-t002:** Specimen information for samples sequenced in this study. Numbers at the end of species names are field codes used to identify individuals. MUSM = Museo de Historia Natural, Universidad Nacional Mayor de San Marcos, UMMZ = University of Michigan Museum of Zoology, EBLA = Estación Biológica Los Amigos, EBMS = Estación Biológica Madre Selva, LBM = Las Brisas del Mogotón, EBVC = Estación Biológica Villa Carmen, RB = Refugio Bartola, mm = millimeters, g = grams, F = female, M = male, J = juvenile, A = adult.

Family	Taxon	Museum Accession No.	Date Captured	Station, Country	SVL (mm)	Mass (g)	Sex	Age
Viperidae	*Bothrops atrox* 0365	MUSM 35721	21 March 2016	EBLA, Peru	589	81	F	J
	*Bothrops bilineatus* 0065	UMMZ 245084	11 March 2016	EBLA, Peru	744	85	F	A
	*Bothrops brazili* 1278	MUSM 36922	1 December 2016	EBLA, Peru	606	76	M	A
Elapidae	*Micrurus annellatus* 3275	UMMZ 248450	26 November 2018	EBLA, Peru	497	18.11	F	A
	*Micrurus hemprichii* 1810	UMMZ 246857	18 January 2017	EBMS, Peru	740	86	M	A
	*Micrurus lemniscatus* 0249	UMMZ 245082	16 March 2016	EBLA, Peru	715	65	M	A
	*Micrurus lemniscatus* 0336	MUSM 35905	21 March 2016	EBLA, Peru	725	50	F	A
	*Micrurus nigrocinctus* 3053	UMMZ 247142	22 May 2018	LBM, Nicaragua	717	64.8	F	A
	*Micrurus obscurus* 0665	UMMZ 246859	7 November 2016	EBVC, Peru	261	5.19	M	J
	*Micrurus obscurus* 1054	UMMZ 246860	22 November 2016	EBLA, Peru	775	81	M	A
Colubridae	*Helicops angulatus* 0143	UMMZ 245053	13 March 2016	EBLA, Peru	373	48.36	F	A
	*Helicops angulatus* 3440	UMMZ 248879	2 December 2018	EBLA, Peru	411	60	F	A
	*Helicops angulatus* 3559	MUSM 39826	9 December 2018	EBLA, Peru	307	24.19	F	A
	*Helicops leopardinus* 1812	UMMZ 246808	18 January 2017	EBMS, Peru	685	220	F	A
	*Helicops polylepis* 1932	UMMZ 246809	18 January 2017	EBMS, Peru	823	600	F	A
	*Leptodeira annulata* 0468	UMMZ 245059	24 March 2016	EBLA, Peru	463	18.56	M	A
	*Leptodeira annulata* 0497	UMMZ 245060	27 March 2016	EBLA, Peru	590	38.02	F	A
	*Leptodeira rhombifera* 3241	UMMZ 247098	12 June 2018	Tecomapa, Nicaragua	665	169.5	F	A
	*Leptodeira septentrionalis* 3176	UMMZ 247099	3 June 2018	RB, Nicaragua	654	113.2	F	A

**Table 3 toxins-14-00489-t003:** Library preparation, sequencing platform, sequencing output, and percent of the transcriptome that was comprised of toxin transcripts for individuals used in the study.

Family	Taxon	Library Preparation	Illumina Platform	Reads Pairs	Percent Toxin Expression
Viperidae	*Bothrops atrox* 0365	TruSeq RNASeq	HiSeq 4000	22,392,182	47.97%
	*Bothrops bilineatus* 0065	NEBNext Ultra II	NovaSeq 6000	17,985,945	74.53%
	*Bothrops brazili* 1278	NEBNext Ultra II	NovaSeq 6000	14,816,998	47.25%
Elapidae	*Micrurus annellatus* 3275	NEBNext Ultra II	NovaSeq 6000	16,398,046	39.12%
	*Micrurus hemprichii* 1810	NEBNext Ultra II	NovaSeq 6000	19,149,986	33.68%
	*Micrurus lemniscatus* 0249	NEBNext Ultra II	NovaSeq 6000	16,413,190	19.58%
	*Micrurus lemniscatus* 0336	TruSeq RNASeq	HiSeq 4000	24,938,732	53.04%
	*Micrurus nigrocinctus* 3053	NEBNext Ultra II	NovaSeq 6000	18,981,692	63.38%
	*Micrurus obscurus* 0665	NEBNext Ultra II	NovaSeq 6000	15,955,904	40.44%
	*Micrurus obscurus* 1054	NEBNext Ultra II	NovaSeq 6000	16,791,601	48.82%
Colubridae	*Helicops angulatus* 0143	TruSeq RNASeq	HiSeq 4000	23,374,958	17.15%
	*Helicops angulatus* 3440	NEBNext Ultra II	NovaSeq 6000	17,274,735	31.03%
	*Helicops angulatus* 3559	NEBNext Ultra II	NovaSeq 6000	15,797,921	19.57%
	*Helicops leopardinus* 1812	NEBNext Ultra II	NovaSeq 6000	17,894,322	52.98%
	*Helicops polylepis* 1932	NEBNext Ultra II	NovaSeq 6000	12,936,858	91.81%
	*Leptodeira annulata* 0468	NEBNext Ultra II	NovaSeq 6000	19,191,193	27.03%
	*Leptodeira annulata* 0497	TruSeq RNASeq	HiSeq 4000	20,844,579	36.83%
	*Leptodeira rhombifera* 3241	NEBNext Ultra II	NovaSeq 6000	17,838,000	41.59%
	*Leptodeira septentrionalis* 3176	NEBNext Ultra II	NovaSeq 6000	17,147,031	46.39%

## Data Availability

The raw sequence data for each individual are available on NCBI SRA under BioProject PRJNA843733 and BioSample accession SAMN28768753-SAMN28768771.

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
