# Peer review of "Divergent Specialization of Simple Venom Gene Profiles among Rear-Fanged Snake Genera (Helicops and Leptodeira, Dipsadinae, Colubridae)"

_toxins, 2022, doi:10.3390/toxins14070489_

Round 1

Reviewer 1 Report

The authors of the paper present the results of describing the overall venom composition and venom diversity of 19 specimens ranging from four different genera. The authors sequenced the venom gland transcriptomes and made claims about the types of toxins and the diversity of toxins found in these different groups. 

Major comments: 

My first comment is that the method section of the paper is not finished. It appears that the authors left out all of section 4.2 about the sequencing methods. This method appears to be a direct copy and paste from the section above. This section needs to be fixed in order to adequately evaluate the paper. 

The next major point is about the database that the authors used to annotate and identify toxins found in their assemblies. Why did the authors only use the selected species and not include the uniprot toxin database for the clade Toxicofera? I feel like this potentially could have biased the toxins that they identified and influenced the diversity counts that the authors display. The authors state that not a lot of information is known about the venom of colubrid snakes and should include some of the other species that they listed in their first table rather than the two species that they selected along with specific class of toxins found in other colubrids. 

The authors state they annotated the transcriptomes using their custom database but they do not describe how they separated their annotations into toxins and nontoxins. Part of the database that they used included the full genome from O. hannah. The authors need to provide more information about this process and the subsequent results in the paper. For example, How were their annotations separated? What is the proportion of the transcriptomes that constituted toxins vs nontoxins? 

In the title and the last line of the abstract the authors mention about divergent specialization and possible rates of venom evolution that could vary across clades. However, nowhere in the paper do the authors test or provide evidence for this. It would be nice to see the authors provide some data on the evolution of the toxin families that they are talking about. The authors have data from multiple lineages and from multiple individuals of the same species. I feel like this paper would better support their claims and their title if they included some sort of selection analysis on top of just describing the venom composition. As it is written the results are very straightforward and provide a description of the venoms but adding a selection analysis to this paper would help to increase the significance of the paper and interest to readers. 

In section 4.4 the authors describe how they calculated the Shannon’s Indices for the different species. I am not entirely sure whether they used all of the toxins and their individual TPM values or the value of the toxin families and their proportion of the toxin transcriptome. Could the author please make this more clear in this section and provide a rationale for why they did it one way or the other? Based on the currently structure it seems like they did it based of the toxin families and not the individual toxins. Why not do this analysis with the individual toxins and their TPM counts? 

Minor Comments

Line 18: I do not understand how the venoms are “streamlined”. Could the authors change this wording or explain a little better in the abstract. 

Line 57: Change to “appear to follow”

Line 262: change to “case in M. obscurus”

Author Response

Response to Reviewers

Thank you very much for reconsidering our manuscript. The referee comments were very helpful in clarifying and strengthening several aspects of the manuscript, especially for clarifying the methods. We believe this revised version is much improved over the original. We thank the referees and editorial team for their time and care in giving thoughtful feedback.

Our principal revisions to this manuscript include clarifications of our methods:

  • Replacing duplicated text with correct paragraph about sequencing methods
  • Rationalizing why we selected specific species as our reference database and how we sorted toxins from non-toxins.
  • A reanalysis of Shannon’s diversity indices.

We have also made minor revisions to phrasing throughout the manuscript to increase clarity

Our responses to specific referee comments are in bold

Referee 1

The authors of the paper present the results of describing the overall venom composition and venom diversity of 19 specimens ranging from four different genera. The authors sequenced the venom gland transcriptomes and made claims about the types of toxins and the diversity of toxins found in these different groups. 

Major comments: 

My first comment is that the method section of the paper is not finished. It appears that the authors left out all of section 4.2 about the sequencing methods. This method appears to be a direct copy and paste from the section above. This section needs to be fixed in order to adequately evaluate the paper. 

Thank you for catching the error!  We have replaced the duplicated text with the correct paragraph about the sequencing methods, revised MS Lines (424-444):

We extracted total RNA from venom glands using the PureLink RNA mini kit (Life Technologies, Carlsbad, CA, USA) following the recommended protocol for animal tissue. Total RNA was stored at -80°C until submission to the University of Michigan’s Advanced Genomics Core where RNA was quantified with a Qubit fluorometer (Invitrogen, Carlsbad, CA, USA) and size was visualized with an Aligent TapeStation (Santa Clara, CA, USA). Poly-A tail selected libraries were constructed with Illumina TruSeq RNASeq (San Diego, CA, USA) and NEBNext Ultra II RNA (Ipswich, MA, USA) library kits and 150bp paired-end sequencing was conducted on Illumina HiSeq4000 or Illumina NovaSeq6000 machines (Table 2).”

The next major point is about the database that the authors used to annotate and identify toxins found in their assemblies. Why did the authors only use the selected species and not include the uniprot toxin database for the clade Toxicofera? I feel like this potentially could have biased the toxins that they identified and influenced the diversity counts that the authors display. The authors state that not a lot of information is known about the venom of colubrid snakes and should include some of the other species that they listed in their first table rather than the two species that they selected along with specific class of toxins found in other colubrids. 

We revised the section to clarify that nontoxin transcripts and toxin transcripts were included in the transcriptomes of the species used in the custom database and clarified why we selected these species. We choose these species as they represent a diversity of snake families and include their whole transcriptome (both toxin and non-toxin transcripts). While uniprot is a useful tool, we find that colubrids are underrepresented in the database (Elapidae = 1080 and Viperidae = 1361 results to Colubridae = 58 and Dipsadidae = 26 results) and thus our combination of species appears to be more robust and representative of colubrid transcripts. Further, a combination of the transcriptomes and genome were used by Modal et al. 2018 when identifying toxins found in the colubrid Spilotes sulphureus, demonstrating precedent for our approach here. We revised MS lines (463-469)

“We created a custom database of nontoxin and toxin nucleotide sequences by downloading venom gland transcriptomes of Crotalus adamanteus [59], Crotalus horridus [73], Micrurus fulvius [74], Boiga irregularis [21], Hypsiglena sp. [21], and Spilotes sulphureus [26], the annotated genome of Ophiophagus hannah [75], and sequences of snake venom matrix metalloproteinase [76] from NCBI GenBank. A combination of these species have been used previously to identify putative rear-fanged snake venom components and represent a diversity of venomous snakes [26].”

The authors state they annotated the transcriptomes using their custom database but they do not describe how they separated their annotations into toxins and nontoxins. Part of the database that they used included the full genome from O. hannah. The authors need to provide more information about this process and the subsequent results in the paper. For example, How were their annotations separated? What is the proportion of the transcriptomes that constituted toxins vs nontoxins? 

We have made revisions to clarify how toxins and nontoxins were separated. We also added a sentence to the results section describing the range of total proportion of the toxin expression found in the transcriptomes and added a column to table 3 with the toxin percentage. We revised MS lines (469-473) in methods section 4.3 and (189-190) in results section 2.2

“We used BLASTn [69] to identify known toxins in our transcriptomes using this custom database. We wrote a custom script in R [77] to annotate our nucleotide sequences, sort non-toxin and toxin transcripts by identifying annotations that matched known toxins from the custom database using the ‘grep’ function, and associate sequence identity with transcript abundance estimates."

In total, we found that total toxin expression encompassed between 17% and 91% of the total expression in the venom gland transcriptomes studied (table 3).

In the title and the last line of the abstract the authors mention about divergent specialization and possible rates of venom evolution that could vary across clades. However, nowhere in the paper do the authors test or provide evidence for this. It would be nice to see the authors provide some data on the evolution of the toxin families that they are talking about. The authors have data from multiple lineages and from multiple individuals of the same species. I feel like this paper would better support their claims and their title if they included some sort of selection analysis on top of just describing the venom composition. As it is written the results are very straightforward and provide a description of the venoms but adding a selection analysis to this paper would help to increase the significance of the paper and interest to readers. 

We agree that a selection analysis is important for understanding the evolution of venoms in the clades we examined. We are currently in the process of analyzing those data for a separate manuscript that specifically focuses on the evolution of toxin gene families. In response to the reviewer’s comment that we do not test for divergent rates of evolution, we revised the phrase in the concluding sentence in the abstract about rates of venom to describe the implications of our results more appropriately (MS lines 17-19).

"These results illustrate the repeatability of simple venom profiles in rear-fanged snakes and relatively constrained nature of venom composition within genera.”

In section 4.4 the authors describe how they calculated the Shannon’s Indices for the different species. I am not entirely sure whether they used all of the toxins and their individual TPM values or the value of the toxin families and their proportion of the toxin transcriptome. Could the author please make this more clear in this section and provide a rationale for why they did it one way or the other? Based on the currently structure it seems like they did it based of the toxin families and not the individual toxins. Why not do this analysis with the individual toxins and their TPM counts? 

 We revised how we calculated Shannon’s Indices as we found analyzing the individual toxins and their TPM’s would yield a more accurate representation of toxin diversity, as done in Holding et al. 2021. Further, we updated figure 2 to contain the revised indices. We revised MS lines (549-551) in methods section 4.4 and (228-231) in results section 2.3

“To estimate levels of venom complexity, we calculated Shannon indices (H’) [94] based on each unique toxin and their respective TPM expressed in transcriptomes of individuals examined.”

“These values ranged from a low of 0.662 for Helicops leopardinus to a high of 3.251 for B. brazili (Fig. 2). Overall, the four genera ranked from lowest to highest levels of venom complexity as follows: Helicops (1.289), Micrurus (1.602), Leptodeira (1.954), and Bothrops (3.014). ”

Minor Comments

Line 18: I do not understand how the venoms are “streamlined”. Could the authors change this wording or explain a little better in the abstract.  

Revised ‘streamlined’ to ‘simple’

Line 57: Change to “appear to follow”

Revised as suggested

Line 262: change to “case in M. obscurus”

Revised as suggested

Reviewer 2 Report

The transcriptomic data on the venom of the rear-fanged Colubridae snake, which has not been well studied, is very significant. It is also interesting to note that the composition of the toxins seems to be dichotomized among the subfamily.

There are many reports that point out a relationship between the diversity and variation of snake venom toxins and diet or environment even within the same species, and similar results have been observed for the snakes in this study.

In this sense, the similarity in the venom profiles of distantly related rear-fanged Leptoderia and front-fanged Bothrops snakes does not seem to make much sense.

Author Response

Response to Reviewers

Thank you very much for reconsidering our manuscript. The referee comments were very helpful in clarifying and strengthening several aspects of the manuscript, especially for clarifying the methods. We believe this revised version is much improved over the original. We thank the referees and editorial team for their time and care in giving thoughtful feedback.

Our principal revisions to this manuscript include clarifications of our methods:

  • Replacing duplicated text with correct paragraph about sequencing methods
  • Rationalizing why we selected specific species as our reference database and how we sorted toxins from non-toxins.
  • A reanalysis of Shannon’s diversity indices.

We have also made minor revisions to phrasing throughout the manuscript to increase clarity

Our responses to specific referee comments are in bold

Referee 2

The transcriptomic data on the venom of the rear-fanged Colubridae snake, which has not been well studied, is very significant. It is also interesting to note that the composition of the toxins seems to be dichotomized among the subfamily.

There are many reports that point out a relationship between the diversity and variation of snake venom toxins and diet or environment even within the same species, and similar results have been observed for the snakes in this study.

In this sense, the similarity in the venom profiles of distantly related rear-fanged Leptoderia and front-fanged Bothrops snakes does not seem to make much sense.

Added clarification in discussion about how this trend is actually common in snakes, as there appears to be four major toxin families used by snake (SVMP, PLA2, 3FTx, and SVSP; Barua and Mikheyev 2019). We revised MS lines (398-402)

Barua and Mikheyev [55] found that while many combinations of venom components were possible, different species tend to show similar venom profiles despite phylogenetic relatedness, albeit with different proportions of respective venom compositions. We find evidence of this trend with the similar venom profiles exhibited by Bothrops and Leptodeira.

Round 2

Reviewer 1 Report

I appreciate the authors thorough response to the problems that I raised during the first review. I think that the manuscript is much clearer. There are just a few more questions and comments that I think could help to improve the manuscript. 

Major Comments:

My first concern is that the authors include the micro-CT scans of the snakes that are in the paper and say that they have deposited this information for public access but do not provide any methods to this part of the paper. The authors need to provide all of the material and methods that they used for this manuscript. 

In the discussion on lines 261 to 263 the authors need to update this discussion to fit with the updated analysis that they have on venom complexity. This section is still referencing the older way that the authors calculated venom complexity and does not encompass the new method that is presented in the paper. This part of the discussion needs to be updated and put into more perspective with the updated methods and results. 

Lines 336 – 339. The authors on these lines talk about the similarities of the profiles for Bothrops and Leptodeira. I feel like this claim is not matched in the graph provided on figure 2. Bothrops has significant differences in abundance of the different groups and even has toxin families at high abundance that are not in Leptodeira.  The authors need to back this claim up with their results or they need to remove this part of the discussion. 

The last major comment that I have is on the results shown in table 3. I appreciated the added percentages of toxins vs nontoxins that they authors have provided. I do feel like part of the discussion should focus on why there is such a large difference in some of these numbers. If the authors are just getting the RNA from the venom gland why are some of the values well below 40%? The few citations that I went through that they authors have of other transcriptomes show a higher percentage of toxin transcripts to non-toxin transcripts. The authors need to address these results in the discussion and provide feedback especially for the really low percentages that are shown in this table. Are the authors missing some of the toxin transcripts from these really low samples?  

The only minor comment is that the authors need to double check all of the references to the tables and figures throughout the paper. Figure 2A is cited in a few spots and does not exist. Section 2.1 references the wrong part of Figure 1 in almost every instance. Lastly section 4.2 references table 2 when it should be table 3. These are the mistakes that I caught but there might be other ones. 

Author Response

Thank you very much for reconsidering our manuscript. Once again, the referee comments were very helpful in clarifying and strengthening several aspects of the manuscript, especially for clarifying the methods and implications in the discussion. We believe this revised version is improved over the original. We thank the referees and editorial team for their time and care in giving thoughtful feedback.

Our principal revisions to this manuscript include clarifications of our methods and expansion of our discussion:

- Addition of our diceCT and segmentation methods.

- Placing Shannon diversity indices in perspective.

- Removal of statements referring to Leptodeira and Bothrops venoms as similar. 

- Addition of possible reasons why toxin transcript abundances varied among specimens examined.

We have also made minor revisions to phrasing throughout the manuscript to increase clarity and corrected issues with figure and table references. We note a discrepancy between the line numbers shown in the .docx and .pdf files. Line numbers mentioned here refer to the .docx file.

Our responses to specific referee comments are in bold

Major Comments:

My first concern is that the authors include the micro-CT scans of the snakes that are in the paper and say that they have deposited this information for public access but do not provide any methods to this part of the paper. The authors need to provide all of the material and methods that they used for this manuscript. 

We have added a new section to the methods describing briefly describing how the specimens were scanned and segmented (MS lines 600-605)

"4.5 diceCT and segmentation

We used diffusible iodine contrast-enhanced computed tomography (diceCT) to scan a representative specimen from each genus using 1.25% Lugol's iodine solution and a Nikon Metrology XTH 225ST microCTscanner (Xtect, Tring, UK) at the UMMZ, following protocols outlined in Callahan et al. [97]. We segmented the maxillary bone and venom gland using the 'draw' and 'thresholding' tools in Volume Graphics Studio Max version 3.2 (Volume Graphics, Heidelberg, Germany)."

In the discussion on lines 261 to 263 the authors need to update this discussion to fit with the updated analysis that they have on venom complexity. This section is still referencing the older way that the authors calculated venom complexity and does not encompass the new method that is presented in the paper. This part of the discussion needs to be updated and put into more perspective with the updated methods and results. 

We added a sentence to the discussion to put the Shannon diversity indices in perspective. We do note that the line numbers provided by the reviewer do not correspond to the appropriate section (lines 261-263 did not exist in the .docx provided). If our correction is not in the appropriate place or does not address the section the reviewer is concerned about, please let us know. We added the following to the discussion (MS lines 349-352)

“Shannon diversity indices suggest that while complexity may vary among individuals, differences may be due to the relative contribution of each underlying transcript rather than differences in toxin family abundance.”

Lines 336 – 339. The authors on these lines talk about the similarities of the profiles for Bothrops and Leptodeira. I feel like this claim is not matched in the graph provided on figure 2. Bothrops has significant differences in abundance of the different groups and even has toxin families at high abundance that are not in Leptodeira.  The authors need to back this claim up with their results or they need to remove this part of the discussion. 

We have decided to remove the statements referring to Leptodeira and Bothrops as similar. Originally, we were referring to the use of SVMP as a dominate toxin but based on the reviewers comments we can see how this claim may feel misleading.

The last major comment that I have is on the results shown in table 3. I appreciated the added percentages of toxins vs nontoxins that they authors have provided. I do feel like part of the discussion should focus on why there is such a large difference in some of these numbers. If the authors are just getting the RNA from the venom gland why are some of the values well below 40%? The few citations that I went through that they authors have of other transcriptomes show a higher percentage of toxin transcripts to non-toxin transcripts. The authors need to address these results in the discussion and provide feedback especially for the really low percentages that are shown in this table. Are the authors missing some of the toxin transcripts from these really low samples?  

We added an explanation for why we think the total proportions of toxin to non-toxins varied. Snakes will highly express toxin genes after a feeding/milking event, then return to a baseline expression level after a period of several days. We were unable to control for this, and as such believe that differences in time from feeding contributed to the variation in total toxin expression. We have added the following sentences to the discussion (MS lines 420-424)

“Further, the proportion of toxin expression in the whole transcriptome varied widely among the specimens examined (17.15%-91.81%; Table 3). The variation in venom expression can possibly be attributed to differences in toxin expression over time [52]. For example, toxin expression is higher after feeding events to replenish the used venom proteins [52].”

The only minor comment is that the authors need to double check all of the references to the tables and figures throughout the paper. Figure 2A is cited in a few spots and does not exist. Section 2.1 references the wrong part of Figure 1 in almost every instance. Lastly section 4.2 references table 2 when it should be table 3. These are the mistakes that I caught but there might be other ones. 

Thank you for bringing this to our attention. We have gone through and corrected the errors and double checked that each reference to a figure or table is now correct.
